# A Functional Analysis of the Unclassified Pro2767Ser BRCA2 Variant Reveals Its Potential Pathogenicity that Acts by Hampering DNA Binding and Homology-Mediated DNA Repair

**DOI:** 10.3390/cancers11101454

**Published:** 2019-09-28

**Authors:** Maria Valeria Esposito, Giuseppina Minopoli, Luciana Esposito, Valeria D’Argenio, Federica Di Maggio, Emanuele Sasso, Massimiliano D’Aiuto, Nicola Zambrano, Francesco Salvatore

**Affiliations:** 1CEINGE-Biotecnologie Avanzate, 8014 Naples, Italy; espositomari@ceinge.unina.it (M.V.E.); giuseppina.minopoli@unina.it (G.M.); dargenio@ceinge.unina.it (V.D.); dimaggio@ceinge.unina.it (F.D.M.); sasso@ceinge.unina.it (E.S.); 2Department of Molecular Medicine and Medical Biotechnologies, University of Naples Federico II, 80131 Naples, Italy; 3Institute of Biostructures and Bioimaging, CNR, Via Mezzocannone 16, I-80134 Naples, Italy; luciana.esposito@unina.it; 4Department of Senology, Istituto Nazionale Tumori—IRCCS Fondazione Pascale, 80131 Naples, Italy; info@daiuto.it

**Keywords:** cancer genomics, breast cancer, mutation, variant of uncertain significance (VUS), VUS classification, DNA damage repair, *BRCA2*

## Abstract

*BRCA1* and *BRCA2* are the genes most frequently associated with hereditary breast and ovarian cancer (HBOC). They are crucial for the maintenance of genome stability, particularly in the homologous recombination-mediated repair pathway of DNA double-strand breaks (HR-DSBR). Widespread *BRCA1/2* next-generation sequencing (NGS) screening has revealed numerous variants of uncertain significance. Assessing the clinical significance of these variants is challenging, particularly regarding the clinical management of patients. Here, we report the functional characterization of the unclassified *BRCA2* c.8299C > T variant, identified in a young breast cancer patient during *BRCA1/2* NGS screening. This variant causes the change of Proline 2767 to Serine in the DNA binding domain (DBD) of the BRCA2 protein, necessary for the loading of RAD51 on ssDNA during the HR-DSBR. Our in silico analysis and 3D-structure modeling predicted that the p.Pro2767Ser substitution is likely to alter the BRCA2 DBD structure and function. Therefore, to evaluate the functional impact of the p.Pro2767Ser variant, we used a minigene encoding a truncated protein that contains the BRCA2 DBD and the nearby nuclear localization sequence. We found that the ectopically expressed truncated protein carrying the normal DBD, which retains the DNA binding function and lacks the central RAD51 binding domain, interferes with endogenous wild-type BRCA2 mediator functions in the HR-DSBR. We also demonstrated that the BRCA2 Pro2767Ser DBD is unable to compete with endogenous BRCA2 DNA binding, thereby suggesting that the p.Pro2767Ser substitution in the full-length protein causes the functional loss of BRCA2. Consequently, our data suggest that the p.Pro2767Ser variant should be considered pathogenic, thus supporting a revision of the ClinVar interpretation. Moreover, our experimental strategy could be a valid method with which to preliminarily evaluate the pathogenicity of the unclassified *BRCA2* germline variants in the DBD and their risk of predisposing to HBOC.

## 1. Introduction

Germline predisposing-mutations in the high-susceptibility *BRCA1* and *BRCA2* genes characterize the hereditary breast and ovarian cancer (HBOC) syndrome (OMIM #604370 and #612555, for *BRCA1* and *BRCA2*, respectively) [1]. This clinical condition is associated with a young onset age, bilateral and aggressive cancers, and a family history of numerous cases of breast and ovarian cancers [2]. Therefore, the identification of *BRCA* mutation carriers is crucial for HBOC prevention, early detection, disease monitoring and also for treatment [3,4].

Routine *BRCA* molecular screening with next-generation sequencing (NGS) techniques is continuously revealing novel variants and variants of uncertain significance (VUSs) [5]. Consequently, the number of unclassified nucleotide changes is increasing. The identification of novel variants and VUSs, particularly at an early age, in relatives of patients and during the screening of non-affected subjects, is a relevant clinical challenge. Indeed, the correct interpretation of the significance of the *BRCA1* and *BRCA2* variants is crucial for genetic counseling and for the clinical management of HBOC patients and their relatives [6,7,8]. Therefore, much effort is being made to determine the role of *BRCA1* and *BRCA2* VUSs and of novel variants [9,10,11,12].

*BRCA1* and *BRCA2* are tumor-suppressor genes involved in a common cell pathway that protects genome integrity [13]. These genes play a relevant role in various steps of the DNA damage response and DNA repair, since they encode proteins intimately involved in cellular growth and differentiation [14]. In particular, by regulating the RAD51 protein activity, the *BRCA2* gene is a crucial mediator of the repair mechanism that is based on homologous recombination (HR) [15,16]. These mechanisms ensure the maintenance of genomic stability by ensuring the correct functionality of the HR pathway in repairing double-strand DNA breaks [17].

In this study, we report the functional characterization of the unclassified p.Pro2767Ser variant in exon 18 of the *BRCA2* gene. This missense variant was identified in a young woman (26 years old at diagnosis) undergoing *BRCA* molecular testing for fibrocystic breast dysplasia. The variant is reported in the ClinVar database (rs587782619) and is classified as a variant of uncertain significance, since its functionality has not yet been described, and since there are no data on its prevalence, nor any information about carriers. We evaluated the functional effect of the p.Pro2767Ser substitution by testing the mutated BRCA2 DNA-binding domain (DBD)together with the nearby nuclear localization sequence (NLS) in in vitro and in living cells assays based on the ability of the mutated or wild-type DBD to compete with the endogenous wild-type BRCA2 activity in HR. We demonstrated, by using an indirect methodological approach, that the BRCA2 p.Pro2767Ser DBD did not affect the endogenous BRCA2 DNA binding, therefore lacking its physiological function.

## 2. Results

### 2.1. The p.Pro2767Ser BRCA2 Variant Identification and Pathogenicity In Silico Predictions

The p.Pro2767Ser BRCA2 (rs587782619) variant was identified in a young woman (26 years old at diagnosis) affected by breast fibrocystic dysplasia, via a *BRCA1/2* NGS screening. Even if this is usually a benign condition, an ultrasound examination showed a highly vascularized hypoechoic area with numerous calcifications that led to a surgical quadrantectomy. Thus, considering this finding and the young age of the patient, she was admitted to a large “Under-forty” surveillance program including a *BRCA1/2* NGS screening. This variant, confirmed on the patient’s DNA by standard Sanger sequencing (Figure 1A), was not found in any of the remaining 849 subjects that were examined, and its allele frequency was 5.8 × 10^−4^%. Molecular testing did not identify any known pathogenic mutation in the *BRCA1* or *BRCA2* genes. The BRCA2 p.Pro2767Ser was annotated on the ClinVar database as a VUS; its prevalence in the general population is unknown.

At the time of the *BRCA* analysis, the family of the patient was uninformative. Due to the limited genetic information available for the family, we conducted both informatics and functional analyses to evaluate whether BRCA2 p.Pro2767Ser could be pathogenic. We performed bioinformatic predictions by using specific software. Both the SIFT and PolyPhen tools classified the variant as “deleterious, with a strong probability of pathogenicity” (Appendix A). Moreover, we found that the 2767 amino acid residue is highly conserved among species, which suggests that it could have a harmful effect. Indeed, it was classified as C65 on the Align-GVGD scale, which ranges from the least likely to be deleterious (C0) to the most likely to be deleterious (C65), namely a variant that probably exerts destructive effects on the phenotype. Furthermore, the Mutation Taster tool [18] classified our variant as “disease causing”, suggesting that protein features might be affected by the nucleotide variation. Moreover, the in silico analysis performed with other bioinformatics tools, including the VarSome website, indicated that p.Pro2767Ser plays a disease-causing role.

### 2.2. 3D Modeling Analysis of the Region Containing the Variant

BRCA2 is a large protein of 3418 amino acids without sequence homology with any other known protein [19]. However, it contains multiple functional domains and interacts with several partners (Figure 1B). The N-terminus of BRCA2 contains the binding sites for the partner and localizers of BRCA2 (PALB2) [20] and eight BRC repeats in the central region that bind RAD51 monomers [21]. The C-terminal region includes the DNA binding domain (DBD), the nuclear localization signal (NLS) [22], and an additional site for RAD51 binding which is regulated by the cyclin-dependent kinase (CDK) phosphorylation of residue Ser3291 [23]. The DBD region contains an N-terminal helical domain (H), followed by three structurally homologous domains, which assume the OB oligonucleotide/oligosaccharide-binding folds (OB1, OB2 and OB3). This region binds single-stranded DNA (ssDNA), whereas a tower-like structure included in the OB2 fold is proposed to bind double-stranded DNA (dsDNA) [24].

There is no available deposited crystal structure of the human BRCA2 protein, but the DBD domain is homologous with the protein from mice (Mus musculus) (sequence identity ∼79%), whose DBD 3D-structure has been determined experimentally. Therefore, a confident structural model for this region can be obtained by homology modeling, using the highest resolution mouse structure (PDB code 1MIU) as a template. Indeed, since the p.Pro2767Ser variant falls in the DBD of BRCA2, we have used the obtained human model to analyze the site of the mutated residue (Figure 2A).

The Pro residue at position 2767 is located in the OB1 domain fold; in particular, it is in the L45 loop, a long stretch connecting strand β4 to β5 of the OB1 fold. The Pro residue, together with the following three residues along the sequence (2767PLEA2770), represents the most conserved region of L45. In the model, the proline residue (p.P2767) is sandwiched between the side chains of M2676 (in α1 of the OB1 fold) and Y2726 (in β3 of the OB1 fold), making van der Waals contacts (<4.0 Å) with them as well as with L2768 in the same loop (Figure 2B). The turn in the loop structure 2766TPLE2769 closes toward the body of its own OB1 domain, at the interface with helix 2617–2637 of the H domain.

The L45 loop structure can be important for both the stability of the OB1 scaffold and the domain-domain interaction between the OB1 and the N-terminal H domain. Indeed, on one hand, within the OB1 fold, the L45 loop acts as a lid by covering a hydrophobic patch (p.V2728, p.M2676, and p.L2776), and it also aids in counterbalancing the nearby positive charges (p.R2678 and p.K2729) by exposing its loop acidic residues (p.D2763, p.E2769, and p.E2772) (Figure 2C). An increased flexibility of the loop resulting in a shift from its packed position could destabilize the OB1 fold by changing its charged and hydrophobic surface properties. On the other hand, the L45 loop structure can be relevant for the correct juxtaposition of the OB1 and H domains. Indeed, p.P2767 interacts with p.Y2726, whose position is crucial since it is hydrogen-bonded to residues from the H domain (p.E2663, p.W2619, and p.H2623). The proper interface between the two domains can also be favored by p.L2768, the Proline adjacent residue, which also packs against another Leu residue (p.L2618) in the facing H domain. It is worth noticing that a correct connection between the two domains (OB1 and N-term H) can be important for the interaction with the DSS1 protein, which in the mouse structure is bounded to a long groove on the surfaces of the OB1 and of the helical domains, mainly at the opposite side of the L45 loop location. The replacement of Pro with a Ser residue may, by increasing the flexibility, significantly destabilize the L45 loop and even the whole protein. Usually, the strong influence of the Proline residue on protein folding and stability is attributed to the limitation of the backbone entropy of unfolding supported by a Proline, which should favor the folding process. The replacement of the rigid Proline with any other more flexible residue may cause a subsequent drop in stability and in some cases even the inability to fold properly.

### 2.3. The p.Pro2767Ser Likely Pathogenic Variant Affects the Binding of BRCA2 to DNA

In mammals, BRCA2 is postulated to be a key mediator in the HR process that acts by promoting the assembly of RAD51 on RPA-coated ssDNA and by allowing the RPA-RAD51 exchange [25]. The C-terminal (CT) region on BRCA2 plays a relevant role in this process. Indeed, in vivo studies showed that the loss of this region causes a hypersensitivity to ionizing radiation, recombination defects and a reduction of the RAD51 foci formation in response to DNA damage [26,27,28,29,30,31,32,33]. These observations suggest that the C-terminal DBD is crucial for the efficient progression of HR.

The BRCA2 p.P2767S likely pathogenic variant located in the DBD could affect the binding of BRCA2 to ssDNA. To explore this issue, we devised an indirect approach by which we analyzed the activity of the endogenous BRCA2 protein in the absence and presence of the ectopic expression of BRCA2 minigenes encoding the isolated CT region. These constructs contain the DNA binding sites, the NLS and one of the RAD51 binding sites (Serine 3291). When ectopically expressed in cells harboring a functional BRCA2 protein, the truncated proteins should affect the endogenous BRCA2 activity in HR by interfering with its binding to DNA.

First, we generated the BRCA2-CT minigene that encodes the FLAG-tagged CT region of the wild-type BRCA2 protein, which ranges from amino acid 2634 to amino acid 3418. Then, we used this construct as a template for two site-directed mutagenesis reactions to reproduce the p.Pro2767Ser mutant DBD (BRCA2-CT P2767S) and the mutant Thr2766AsnFs (BRCA2-CT T2766NFs) (Figure 3A) that is reported in ClinVar as pathogenic (c.8297delC, rs80359705) and used it as a positive control. Indeed, this deletion gives rise to a premature stop codon that results in the loss of about 900 amino acids at the CT domain of the BRCA2 protein (Figure 3A,B). Both mutagenesis reactions were confirmed by Sanger sequencing (Appendix A). Moreover, we site-directed mutagenized 7 missense variants chosen by Guidugli et al. [34] as a validation set: BRCA2-CT L2688P, BRCA2-CT T2722R, BRCA2-CT D2723H, BRCA2-CT D2723G, and BRCA2-CT G2748D as pathogenic controls; BRCA2-CT A2717S and BRCA2-CT K2729N as benign controls.

The fundamental assumption for the validity of our assays is that the truncated constructs are properly expressed and that they are able to enter the nucleus. Therefore, a western blot analysis (Figure 3C and Appendix A) was used to assess whether proteins were properly expressed, and immunofluorescence (Figure 3D) was used to assess the proper expression and localization of the BRCA2-Ct minigenes and whether the NLS was able to transport into the nucleus (see Methods) in murine NIH-3T3 and human U2OS cell lines. No differences between WT and pathogenic variants were present, suggesting that their expression and nuclear localization were comparable and thus do not affect the subsequent analytical procedures.

We determined the effects of the wild-type BRCA2 minigene (BRCA2-CT wt) and of the mutant minigenes (BRCA2-CT P2767S, BRCA2-CT T2766NFs and BRCA2-CT CTRs 1–7) using an in vitro ssDNA-binding assay. Nuclear extracts from 10 Gγ non-transfected irradiated (IR) and non-transfected non-irradiated (NT) cells were incubated with 100 nt-long Cy5-labelled ϕX174 ssDNA, and the protein-DNA complexes were analyzed by electrophoresis (Appendix A). We showed that the non-transfected irradiated cells (Appendix A, lane 3) had a delayed electrophoretic migration with respect to the non-transfected and non-irradiated cells (Appendix A, lane 2). In addition, we tested the transfected NIH-3T3 cells without IR (Appendix A), observing that all samples showed the same behavior. Interestingly, all the BRCA2-CT minigenes showed a slow band migration, including the BRCA2-CT WT-transfected cells that behaved differently from the IR samples. This suggests that the damage-sensitive proteins were also activated by transfection procedures, such as by the serum stimulation or by cell replication mechanisms. Moreover, the ssDNA-binding assay was performed using an increasing amount of the nuclear extract of the irradiated NIH-3T3 cells transfected with the BRCA2-CT L2688P and K2729N minigenes: the DNA-protein complex formation increased when more protein was present (Appendix A). Differences among the minigenes became clear at 5 μg of nuclear extract (Appendix A, lanes 5).

To assess the effect of the BRCA2-CT wt and of the mutant minigenes (BRCA2-CT P2767S, BRCA2-CT T2766NFs and BRCA2-CT CTRs 1–7), we incubated nuclear extracts from 10 Gγ irradiated mock-transfected cells and cells expressing the BRCA2-CT wt, BRCA2-CT P2767S, and BRCA2-CT T2766NFs proteins with the Cy5-labelled ϕX174 ssDNA. In the mock-transfected sample, the endogenous protein complexes bound to ssDNA (Figure 4A, lane 1). The nuclear extracts from the BRCA2-CT P2767S and BRCA2-CT T2766NFs samples (Figure 4A, lanes 3 and 4) had the same intensity as the shifted complexes, similar to the mock-transfected cells, whereas the nuclear extracts from the cells transfected with BRCA2-CT wt affected the ssDNA-protein complexes of the irradiated cells (Figure 4A, lane 2). These results show that the BRCA2-CT wt protein interfered with the assembly of endogenous protein complexes that were activated upon DNA damage and consequently hampered the binding of BRCA2-RAD51 complexes to ssDNA. Instead, the ectopically expressed mutated BRCA2-CT T2766NFs protein, which is known to be unable to bind DNA, did not interfere with the endogenous BRCA2 binding. Similarly, the BRCA2-CT P2767S mutant enabled the proper binding of endogenous proteins. The interference of the BRCA2-CT wt protein could be due to competition with the endogenous BRCA2 protein for ssDNA binding.

Furthermore, the nuclear extracts from the BRCA2-CT WT and from the BRCA2-CT P2767S were compared with all BRCA2-CT CTRs 1–7. (Figure 4B,C). Interestingly, our tested BRCA2-CT P2767S sample (Figure 4B, lane 4 and Figure 4C, lane 3) showed a shift pattern similar to the mock-transfected cells (Figure 4B, lane 2) and to all the 5 pathogenic BRCA2-CT controls (Figure 4B, lanes 5, 6 and 7, and Figure 4C, lanes 6 and 7). Finally, the shift pattern of the 2 tested benign controls (Figure 4C, lanes 4 and 5) was similar to the BRCA2-CT WT. The free probe was loaded in lane 5 of the first gel (Figure 4A) and in the first lane of Figure 4B,C, representing the entire reaction in which the 100 nt-long Cy5-labelled ϕX174 ssDNA was without protein nuclear extract.

The DNA binding specificity is critical for the association of RAD51 with ssDNA [35]. Therefore, because of their inability to interact with DNA and/or with proteins involved in this repair step, the BRCA2-CT P2767S, BRCA2-CT T2766NFs and BRCA2-CT pathogenic CTRs 1–7 mutants could not interfere with the endogenous BRCA2 functionality that leads to the band shift of the endogenous BRCA2 protein.

To investigate the competition of the BRCA2-CT minigenes with the endogenous BRCA2 protein, we depleted the endogenous BRCA2 by using the BRCA2-3-unique 27mer siRNA duplexes. We performed the BRCA2 silencing in the NIH-3T3 cells. Forty-eight hours after the siRNA and BRCA2-CT minigenes co-transfection, the cells were irradiated, and they were harvested 4 h later to isolate the nuclear proteins. The effects of the endogenous BRCA2 silencing on our BRCA2-CT minigenes were assessed by performing an in vitro ssDNA-binding assay in the same conditions as those described above. The nuclear extracts from the 10 Gγ-irradiated and BRCA2-silenced NIH-3T3 cells were incubated with the 100 nt-long Cy5-labelled ϕX174 ssDNA, and the protein-DNA complexes were analyzed by electrophoresis (Figure 4D). Interestingly, in cells where endogenous BRCA2 was silenced, there were no differences of migration patterns among the samples (Figure 4D, lanes 3, 5 and 7). Instead, in the non-silenced cells, the BRCA2-CT minigenes showed different shift patterns according to their capability to compete with endogenous BRCA2 and to bind the ϕX174 ssDNA probe (Figure 4D, lanes 2, 4 and 6).

These results suggest that, unlike BRCA2-CT wt, BRCA2-CT P2767S and BRCA2-CT T2766NFs do not bind broken DNA in living cells. To verify this hypothesis, we used a previously described protocol [36] that permeabilizes cells using detergents before fixation to displace proteins that are not bound to chromatin from the nucleus (Figure 5). U2OS cells were transfected with the wild-type and mutant minigenes. Twenty-four hours after the transfection, the cells were treated with a high dose of X-ray (10 Gy) and fixed four hours later. Before immunofluorescence staining, the cells were fixed without treatment or were permeabilized before fixation to wash away proteins that were not bound to chromatin, as described in the Methods section. The confocal microscopy of the U2OS cells revealed that the BRCA2-CT wt, BRCA2-CT P2767S and BRCA2-CT T2766NFs flag-tagged proteins localize in the nuclei of non-permeabilized cells (Figure 5B–D). This demonstrates that the inability of mutant proteins to interfere with the endogenous proteins was not a consequence of their absence in the nucleus. In permeabilized cells, RAD51, which was bound to chromatin, was visible (Figure 5F–H), but only the BRCA2-CT wt protein was detected, whereas the BRCA2-CT P2767S protein and, notably, the BRCA2-CT T2766NFs protein were washed out, as expected for proteins that do not bind the DNA.

In addition, to assess the BRCA2-CT proteins localizations, we quantified the percentage of transfected cells with and without permeabilization. Taking into account that permeabilized cells derived from slides of the same transfection plate (see Material and Methods), for each BRCA2-CT sample, the total number of FLAG-expressed cells before the permeabilization procedures was used as the “absolute value” for counting. We noticed that, without permeabilization, the BRCA2-CT WT and the BRCA2-CT P2767S were predominantly localized in the nucleus, while the BRCA2-CT T2766NFs was principally in the cytoplasm (Figure 5I). Instead, after the permeabilization procedure, we detected 83% of the BRCA2-CT WT protein in the nucleus, while only 13% of the BRCA2-CT P2767S was found in the nucleus, and no BRCA2-CT T2766NFs was detected in either the nucleus or in the cytoplasm (Figure 5L). In Figure 5I,L, the Y-axis represents the percentage of BRCA2-CT minigenes that localize predominantly in both the nucleus and the cytosol or only in the cytosol.

Thus, the ssDNA binding assay and IF confirm, in two different systems, both the ability of the BRCA2-CT WT and the failure of the pathogenic mutated BRCA2-CTs to bind the DNA. Taken together, the above results indicate that the p.P2767S likely pathogenic variant hampers the binding of BRCA2 DBD to DNA.

### 2.4. The p.Pro2767Ser Likley Pathogenic Variant Affects RAD51 Foci Formation and Repair of DNA Damage

To evaluate the effects of BRCA2-CT P2767S on the HR process, we analyzed the RAD51 foci formation in U2OS cells by immunofluorescence at 0, 30 min, 1 h, and 4 h after irradiation. Four hours was deemed to be the best point at which to analyze and count the RAD51 foci in the cell nuclei. As shown in Figure 6A, which is only representative of several acquisitions on which RAD51 foci were counted, RAD51 foci accumulated in the cell nuclei, which demonstrates that the DNA damage repair pathway was activated after irradiation.

The RAD51 foci were counted in the transfected cells. All the BRCA2-CT minigenes allowed for BRCA2 foci formation following IR, even if in different amounts. The number of foci was fewer in the BRCA2-CT wt nuclei than in BRCA2-CT P2767S (*p* ≤ 0.001) and the mock-transfected nuclei (*p* ≤ 0.01), confirming the wt competition with the endogenous BRCA2 (Figure 6B). In particular, a comparable reduction of the RAD51 foci intensity was noticed in all the BRCA2-CT transfected cells, probably due to BRCA2-CT transfection effects; indeed, the mock-transfected cells showed more foci. However, the reduction of the RAD51 foci intensity of cells expressing the BRCA2-T2766NFs is lower than for the BRCA2-CT WT-transfected cells but greater than for the BRCA2-CT P2767S and mock-transfected cells. These results support the hypothesis that BRCA2-CT wt prevents the recruitment of RAD51 at the sites of damaged DNA. Conversely, BRCA2-CT P2767S and BRCA2-CT T2766NFs did not interfere with the RAD51 foci formation, thus confirming again the substantial different behavior with BRCA2-CT WT in DNA binding and repair.

To evaluate the ability of p.Pro2767Ser to negatively affect the DNA repair pathway, we analyzed the recovery from DNA damage of NIH-3T3 cells transfected with BRCA2-CT wt, with the BRCA2-CT P2767S and with the BRCA2-CT T2766NFs minigenes, and co-transfected with a GFP expression vector (Figure 7A). After irradiation, the percentage of phospho-H2AX-positive cells was counted and normalized on green fluorescent protein (GFP) levels (Figure 7B). At time 0, all non-treated (NT) cells had basal levels of histone H2AX phosphorylation. Thirty minutes after gamma irradiation, all samples had high levels of H2AX phosphorylation, thereby confirming the occurrence of the DNA double-strand break. The percentage of γ-H2AX-positive cells peaked after 12 h, and the level of phosphorylation started to decrease at 24 h in the mock-transfected cells. Interestingly, the DNA of the cells transfected with the BRCA2-CT wt minigene were not repaired (>90% nuclei were phospho-H2AX-positive), whereas the decrease of γ-H2AX-positive signals in the cells transfected with the BRCA2-CT P2767S and BRCA2-CT T2766NFs minigenes was the same as for the mock-transfected cells. Indeed, at 36 h post-IR, the percentage of γ-H2AX was significantly higher in BRCA2-CT wt (83.9%, *p* ≤ 0.0001) than in BRCA2-CT P2767S (16.5%, *p* ≤ 0.0001) or in BRCA2-CT T2766NFs (11.6%, *p* ≤ 0.0001) and the mock-transfected cells (11%, *p* ≤ 0.001), which demonstrates that proteins affected the repair pathway differently and that only BRCA2-CT wt prevented the repair mechanisms (see Figure 7’s legend for details). Therefore, the differences between BRCA2-CT P2767S, BRCA2-CT T2766NFs and the mock-transfected samples were not statistically significant.

Moreover, to further validate our results, we performed the same DNA damage recovery assay for BRCA2-CT wt, BRCA2-CT P2767S and also for all the BRCA2-CT CTRs 1–7 minigenes. After 5Gγ irradiation, the percentage of phospho-H2AX-positive cells was counted at different times (Figure 8).

Interestingly, the DNA of the cells transfected with the BRCA2-CT wt minigene and the BRCA2-CT benign controls (BRCA2-CT A2717S and BRCA2-CT K2729N) was not repaired (> 98% of the nuclei were phospho-H2AX-positive for BRCA2-CT WT; 95% and 93% positive nuclei for BRCA2-CT A2717S and BRCA2-CT K2729N, respectively). On the contrary, the decrease of γ-H2AX-positive signals in the cells transfected with BRCA2-CT P2767S and all the BRCA2-CT pathogenic controls minigenes (BRCA2-CT L2688P, BRCA2-CT T2722R, BRCA2-CT D2723H, BRCA2-CT D2723G, and BRCA2-CT G2748D) were similar to the mock-transfected cells, although the speed of repair varied slightly between the different BRCA2-CT control samples (Figure 8).

In detail, at 36 h post-IR, the percentage of γ-H2AX was significantly higher in BRCA2-CT wt (90.6%, *p* ≤ 0.0001) than in BRCA2-CT P2767S (27.76%, *p* ≤ 0.0001), BRCA2-CT L2688P (17.14%), BRCA2-CT T2722R (11.43%), BRCA2-CT D2723H (10.81%), BRCA2-CT D2723G (12%), BRCA2-CT G2748D (totally repaired, 0%), and the mock-transfected cells (8.47%). All these data demonstrate that mutagenized proteins interfere differently with the repair pathway and that both the BRCA2-CT wt and the BRCA2-CT benign controls prevent repair mechanisms. Indeed, the BRCA2-CT A2717S and BRCA2-CT K2729N-transfected cells showed high levels of γ-H2AX positive cells (87.50% and 85.11%, *p* ≤ 0.0001).

## 3. Discussion

The classification of missense germline variants in the *BRCA1/2* genes as pathogenic could have a significant impact on the evaluation of cancer risk. Consequently, screening for their early detection may be relevant in the clinical management of carriers and in cancer prevention [37]. The introduction of molecular screening by Next-Generation Sequencing has revealed a large number of *BRCA1/2* VUSs. Currently, classification of these VUSs as pathogenic involves the methods and criteria indicated by IARC [38] and ENIGMA [39], which are based on genetic association studies, posterior-probabilities analysis and functional studies. Although these criteria are widely accepted, the classification of VUSs is still a challenge due to several factors (i.e., their low frequency, the lack of family information, and the difficulty in performing functional studies due to the large size of proteins). Moreover, in silico approaches alone are neither sufficiently sensitive nor specific. In addition, the methods currently used to determine VUS pathogenicity require BRCA2-null or conditional knockout cells and the use of a full-length protein. Guidugli et al. [34], performed an extensive review of the functional assays currently available for testing the pathogenic effect of missense *BRCA2* variants, highlighting the need to functionally assess the impact of VUSs [34,40,41,42,43,44]. However, all the described methods are cumbersome and complex; in this context, the method we report here overcomes the aforementioned limitations.

The unclassified BRCA2 p.Pro2767Ser variant, which we identified in a young woman with clinically suspected HBOC, lies in the DBD of BRCA2. The ClinVar database notes the variant as a VUS (rs587782619). However, no information about its prevalence or about the availability of functional assays to assess its clinical value is listed.

Although analyses of hundreds and sometimes thousands of variants are commonly reported, we have paid attention to only one VUS because of different reasons. The first is that this variant was novel when it was found in a young woman who underwent *BRCA1/2* NGS screening. Given the young age of the patient and since no other pathogenic mutations were found, we decided to analyze this variant in order to better characterize its role in disease onset, and in order to allow clinicians to better manage counseling, any therapies, follow-up, and, principally, a cancer-risk assessment in relatives. In addition, we described a novel methodology to assess the role of variants that fall in the DBD of BRCA2, useful for overcoming the complex assays for which the entire BRCA2 protein is required; this method, indeed, allows one to study the role of BRCA2 VUS more quickly than a “traditional” functional assay would, allowing for faster clinical decisions for patient health.

To characterize the p.Pro2767Ser variant, we developed a functional assay in which we analyzed the effects of the overexpression of a minigene encoding the CT region of the BRCA2 mutant protein on the functionality of the endogenous BRCA2 protein. Consequently, given the absence of important functional domains in the N-terminal region, any effect induced by the minigene-encoded protein would be due exclusively to the activity of the DBD, the NLS and other domains in the CT region.

Using this method, we indirectly demonstrated that the BRCA2-CT P2767S protein, similarly to the BRCA2-CT T2766NFs protein, does not affect the endogenous BRCA2 DNA repair machinery because nucleotide substitutions in sequences encoding the DNA binding domains prevent them from interacting with broken ssDNA.

These results correlate with bioinformatics predictions that strongly supported p.Pro2767Ser pathogenicity. First, the p.P2767 residue is highly conserved among species, which suggests that it plays a fundamental functional role. Second, the substitution of proline with serine generates a polarity and modifies the hydrophobicity, thereby altering the protein structure. The latter structural modifications may disrupt external interactions with other molecules on the protein surface. Moreover, this amino acid change can compromise the DNA binding of the BRCA2 protein and, consequently, also compromise the repair efficiency. In this context, we can hypothesize that the p.Pro2767Ser variant impairs the correct folding of BRCA2 or, at least, by increasing the flexibility of the L45 loop, that it affects the fold of the OB1 domain, an essential module of the BRCA2 multi-domain protein. It is worth noting that these considerations still hold for the truncated mutated protein (BRCA2-CT P2767S), even though it does not contain the full N-terminal helical domain.

On the other hand, we can surmise that the wild-type truncated protein (BRCA2-CT wt) with the unmodified loop contains the intact structure of the three OB domains and can thus interfere with the endogenous wt full-length BRCA2 functions in the homologous recombination of damaged DNA. Indeed, the current available information on BRCA2 ssDNA binding, derived from biological data as well as from X-ray structures, indicates that the main site for ssDNA is the tandem OB2-OB3 fold. However, it has been observed that the BRCA2 fragment containing only the helical domain and the OB1 domain also has ssDNA binding activity, albeit much weaker that the intact DBD domain. Based on a recent biochemical and electron microscopy structural investigation of full-length BRCA2, it has been recently proposed that both ssDNA binding sites simultaneously bind to a long ssDNA stretch (> 50 nt), each site on a different monomer, so that the BRCA2 binds the DNA as a dimer [26]. This finding supports the importance of all three OB domains for DNA binding. We also cannot rule out the possibility that, upon dimer formation, a rearrangement occurs of the relative positions of the OB domains, with loop L45 playing a role in this. In conclusion, the BRCA2-CT P2767S presumably does not retain the correct folded structure of the three successively packed OB domains; hence, it is not able to bind the ssDNA. On the other hand, BRCA2-CT wt is able to bind the DNA, but the complex thus formed cannot evolve further. Indeed, the truncated protein lacks other essential domains for the biological function of BRCA2 in the repair of DNA double-strand breaks and inter-strand cross-links by RAD51-mediated homologous recombination [27]. Therefore, BRCA2-CT wt, by binding and sequestering ssDNA, is able to interfere with the activity of the endogenous BRCA2, whereas BRCA2-CT P2767S has no effect on it.

Our ssDNA in vitro results support the hypothesis that the minigene BRCA2-CT wt hampers the endogenous BRCA2 functionality because of the lack of a great part of the protein at the middle and N-terminal regions, and that it prevents the interaction between the endogenous BRCA2 and other DNA binding proteins on double-strand breaks, such as RAD51.

To further verify whether BRCA2-CT wt hampers the repair machinery, we evaluated the DNA damage recovery by counting the phospho-H2AX-positive cells. Indeed, H2AX signaling, which is immediately activated upon damage to double-strand DNA, tends to disappear as the DNA is repaired [36,45]. Therefore, we found that only cells transfected with the BRCA2-CT wt protein and the 2 BRCA2-CT A2717S and BRCA2-CT K2729N benign controls exerted a competitive effect on the capability of the endogenous BRCA2 protein to bind DNA.

These findings converge on the hypothesis that BRCA2-CT P2767S is not capable of binding DNA.

Last, although not working with the full-length recombinant protein, we clearly showed that the behavior of the p.Pro2767Ser protein differs from that of the wild-type protein, thereby reinforcing the concept that this VUS could contain a harmful variant and could be considered clinically relevant. Moreover, using our system, we were able to overcome the lack of a specific model with which to predict the effect of a nucleotide variation in either *BRCA1* or *BRCA2*. However, the use of partial proteins could represent a limitation, since this assay can only be used against a known protein-protein interaction of BRCA2, or for the interaction of these protein complexes with DNA, and may potentially overlook other unidentified interactions. However, this approach could be an opportunity to rapidly assess if a DBD-VUS is potentially pathogenic in order to make clinical decisions and to improve the clinical management of patients and their relatives. On the other hand, another limitation of working with partial protein is that we cannot fully predict the behavior of the variant on the full-length protein, and this is the reason why family information is important for a *BRCA2* VUS evaluation. In this context, a limitation of this study was that we were not able to test other genes besides *BRCA1/2* and that no data about family segregation were available, because family members refused to undergo genetic tests.

For this reason, it is important to maintain a certain caution in the management of this variant, although it should be considered as being likely pathogenic.

## 4. Materials and Methods

### 4.1. Patient and BRCA NGS Screening

The carrier of the *BRCA2* variant attended the Breast Unit of the “Istituto Nazionale dei Tumori-Fondazione G. Pascale”, in Naples, Italy. She underwent breast surgery for fibrocystic dysplasia at age 26. After genetic counseling according to the National Comprehensive Cancer Network (NCCN, http://www.nccn.org) guidelines for HBOC genetic risk assessment [46], she underwent *BRCA* NGS mutation screening, as described elsewhere [47]. The patient was fully informed about the study and provided written informed consent before the blood sample collection. The study was carried out according to the tenets of the Helsinki Declaration and approved by the Istituto Nazionale Tumori-Fondazione G. Pascale Ethics Committee (protocol number 3, dated 25 March 2009). However, an assessment of the functional role of the variant was performed on the patient’s DNA sequence and not directly on her biological samples. Her relatives were not available for a molecular analysis.

### 4.2. Gene, Nomenclature and Databases

P.Pro2767Ser (c.8299C > T, p.Pro2767Ser) is located on the *BRCA2* gene (RefSeq: NM_000059.3, NG_012772.1, NC_000013.11, Ensembl: ENSG00000139618, GenBank: U43746.1). The nomenclature of the DNA variant is based on the *BRCA2* cDNA sequence (Ensembl: ENST00000380152), according to the recommendations of the Human Genome Variation Society (HGVS, http://www.hgvs.org/). The variant is reported in the National Center for Biotechnology Information (http://www.ncbi.nlm.nih.gov/) and in the ClinVar databases as rs587782619 and is classified as a variant of unknown clinical significance; no data regarding its allele frequency are available.

### 4.3. Bioinformatics and 3D Modeling

Bioinformatics predictions of the variant’s effects were performed using the SIFT (http://sift.jcvi.org/) and PolyPhen-2 (http://genetics.bwh.harvard.edu/pph2/) tools. The evolutionary conservation of the variant was analyzed using Align-GVGD (http://www.agvgd.iarc.fr/). Further predictions were assessed with the Mutation Taster tool (http://www.mutationtaster.org) [18], VarSome (https://varsome.com/variant/hg19/chr13-32937638-c-t) and dbNSFP [48]. All software were used with the default parameters.

The 3D modeling of the structure was performed using the Modeller software [49]. A template structure for the homology modeling was derived from the Protein Data Bank (PDB, https://www.rcsb.org). The highest resolution mouse structure that was available (PDB code 1MIU) was used as a template for building the 3D models for the DNA binding domain (DBD: residues 2479–3188; N-terminal helical domain and OB1 domain: residues 2479–2808). The figures of the structural models were generated via the PYMOL software [50].

### 4.4. Cloning and Site-Directed Mutagenesis

The cDNA fragment encoding the C-terminal region of BRCA2 (BRCA2-CT), which contains the DNA binding sites, the nuclear localization sequence and the RAD51 binding regions in the C-terminus encoded by exon 27 [30], was amplified by PCR from the pcDNA3 236HSC WT expression vector (Addgene, Cambridge, MA, USA) [51] using oligonucleotides a and b, as shown in Appendix A, and cloned in the NotI-BamHI sites of the p3XFLAG-CMV™ vector (Sigma-Aldrich, Saint Louis, MO, USA). The c.8299C > T nucleotide change and the c.8297delC deletion were obtained by the in vitro mutagenesis of p3XFLAG-CMV-BRCA2 cDNA using the QuikChange II XL Site-Directed Mutagenesis Kit (Agilent Technologies, Santa Clara, CA, USA), according to the manufacturer’s instructions. Sanger sequencing was performed to verify all of the mutagenized DNA constructs. Seven missense variants, falling in the OB1 DBD of BRCA2, were used to validate the method: 5 variants were classified as pathogenic, namely: (1) p.Leu2688Pro c.8063T > C, rs80359045; (2) p.Thr2722Arg c.8165C > G, rs80359062; (3) p.Asp2723His c.8167G > C, rs41293511; (4) p.Asp2723Gly c.8168A > G, rs41293513; and (5) p.Gly2748Asp c.8243G > A, rs80359071; and 2 variants were classified as being benign/likely benign, namely: (6) p.Ala2717Ser c.8149G > T, rs28897747 and (7) p.Lys2729Asn c.8187G > T, rs80359065 [34].

### 4.5. Cell Cultures and Transfections

NIH-3T3 mouse fibroblasts, obtained from the CEINGE-Biotecnologie Avanzate Cell Bank (Naples, Italy), were grown in Dulbecco’s modified Eagle medium (DMEM, Sigma-Aldrich) supplemented with 10% Fetal Bovine Serum (Sigma-Aldrich), 2 mM L-Ultraglutamine 1 (Lonza, Basel, Switzerland), and 1% penicillin and streptomycin (100 μg/mL streptomycin and 100 units/mL penicillin) (Sigma-Aldrich), at 37 °C with 5% CO_2_. The transfections were done with the indicated cDNAs using Lipofectamine 2000 (Invitrogen, Carlsbad, CA, USA), according to the manufacturer’s instructions. These cells show a high transfection efficiency, which makes them very useful for BRCA2 protein expression studies [52].

The osteosarcoma cell line U20S, also obtained from the CEINGE Cell Bank, was cultured in Dulbecco’s modified Eagle medium (DMEM, Sigma-Aldrich) supplemented with antibiotics, 10% fetal bovine serum (Sigma-Aldrich) and 2 mM L-Ultraglutamine 1 (Lonza), at 37 °C under a humidified 5% CO_2_ atmosphere. The U2OS cells were co-transfected with the p3xFLAG-CMV-BRCA2 cDNAs and the pmaxGFP^®^ Vector (Lonza) (10:1 rate) using the Cell Line Nucleofector^®^ Kit V (Lonza), according to the manufacturer’s protocol. The cell vitality and transfection efficiency were analyzed 24 h post-electroporation using fluorescence microscopy.

### 4.6. Protein Extracts and Western Blot Analysis

Whole-cell protein extracts and nuclear protein extracts were used for the western blot analysis. To prepare these extracts, 1 × 10^7^ cells were harvested in cold phosphate-buffered saline at about 30 h post-transfection, and sonicated in a RIPA buffer (25 mM Tris-HCl pH 7.6, 150 mM NaCl, 1% NP-40, 1% sodium deoxycholate, and 0.1% SDS) (ThermoFisher Scientific, Waltham, MA, USA) supplemented with a protease inhibitor cocktail (104 mM AEBSF, 80 μM Aprotinin, 4 mM Bestatin, 1 mM × 10^−64^, 1.4 mM leupeptin and 1.5 mM pepstatin A) (Sigma-Aldrich). The extracts were clarified by centrifugation at 30,000× *g* at 4 °C, and the protein concentration was determined with the Bio-Rad protein assay (Bio-Rad, Hercules, CA, USA) according to the manufacturer’s instructions.

The nuclear extracts were prepared using 400 μL of ice-cold lysis buffer (10 mM Hepes pH 8.0, 60 mM KCl, 0.1 mM EDTA, 0.2% NP-40, 1 mM DTT, 1 mM Na orthovanadate, 1 mM PMSF, 0.5 M Na fluoride and a protease inhibitor cocktail) and left on ice for 5 min. The supernatant was separated from the nuclear pellet by centrifugation at 1500× *g* at 4 °C for 5 min. The nuclear pellet was washed with lysis buffer without NP-40 and was centrifuged at 1500× *g* at 4 °C for 5 min. The supernatant was discarded, and the nuclei pellet was suspended in 100 μL of lysis buffer without NP-40 and overlaid on 300 μL of a sucrose cushion solution (10 mM Hepes pH 8.0, 60 mM KCl, 0.1 mM EDTA, 30% sucrose, 1 mM DTT, 1 mM Na orthovanadate, 1 mM PMSF, 0.5 M Na fluoride and a protease inhibitor cocktail). The nuclei were differentially centrifuged at 5000× *g* at 4 °C for 20 min, and the supernatant was discarded. The nuclei pellet was suspended in 130 μL of nuclei lysis buffer (TRIS pH 7.5, 60 mM KCl, 1 mM DTT, 1 mM Na orthovanadate, 1 mM PMSF, 0.5 M Na fluoride and a protease inhibitor cocktail) and subjected to 3 cycles of freezing/thawing. The nuclear extracts were clarified by centrifugation for 20 min at 5000× *g* at 4 °C, and the supernatant was quantified using the Bradford assay (Bio-Rad).

The proteins (50 μg) were separated on 4–15% Mini-PROTEAN TGX precast gel (Bio-Rad) and transferred onto a polyvinylidene difluoride Immobilon P membrane (Bio-Rad) using the Trans-Blot Turbo Transfer System (Bio-Rad). Filters were blocked with 3% milk-TBST (50 mMTris pH 7.5, 150 mM NaCl, and 0.1% Triton X-100) for 1 h at room temperature. The BRCA2 mutant proteins were immunodetected using the Monoclonal ANTI-FLAG^®^ M2, Clone M2 (F1804, Sigma-Aldrich). Actin, used to normalize the amount of protein in different samples, was revealed using an anti-β-actin mouse primary antibody (Sigma-Aldrich, 1:1000). Anti-mouse IgG conjugated to horseradish peroxidase was used for the secondary antibodies (GE Healthcare, Chicago, IL, USA). The bands were detected with ECL chemiluminescence methods (PerkinElmer, Waltham, MA, USA) and by exposure to an X-ray film (Fujifilm, Tokyo, Japan) or with the ChemiDoc MP System (Bio-Rad).

### 4.7. Cell Irradiation

The NIH-3T3 cells were irradiated with 6-MV X-rays from a RS-2000 irradiator (RadSource Technologies Inc., Buford, GA, USA) at a dose rate from 5 to 10 Gy. The U2OS cells were irradiated with 10 Gy.

### 4.8. In Vitro ssDNA Binding Assay

The NIH-3T3 cells were transfected with 3XFLAG-BRCA2 cDNAs and irradiated with 10 Gy 24 h later. Four hours after irradiation, the cells were harvested, and the nuclear extracts were prepared as described above. For the in vitro ssDNA binding assay, a linear 100-mer ϕX174 single-stranded DNA (Bio-Fab Research, RM, Italy, Appendix A) was used to determine the binding of wild-type and mutant BRCA2 proteins to DNA. The reactions were prepared by incubating 5 μg of nuclear extracts from the transfected NIH3T3 cells in 50 mM TRIS pH 7.5, 60 mM KCl, 2 mM CaCl_2_, 1 mM ATP, 1 mM DTT, 1 mM MgCl_2_, 0.05 mg/mL BSA and 2 μg Poly(dI-dC) double-strand (Amersham Biosciences, Little Chalfont, UK) for 15 min at 37 °C. Then, 3 pmol of linear ϕX174 5′-Cy5-end-labeled ssDNA oligonucleotides were added, and the reaction was incubated at 37 °C for an additional 45 min. The reactions were stopped by adding glycerol at a final concentration of 36%. The protein-DNA complexes were resolved in 0.6% agarose gel with 0.25× Tris-Borate-EDTA buffer. The gels were run for 3 h at 40 mV at 4 °C. The protein-DNA complexes were detected using a Typhoon FLA scanner (GE Healthcare).

### 4.9. Endogenous BRCA2 Silencing

The endogenous BRCA2 silencing was performed by using the Brca2 (Mouse)—3 unique 27mer siRNA duplexes (SR423642, OriGene Technologies, MD, USA) in the NIH-3T3 cells.

Briefly, 1 × 10^5^ cells were plated on 6-well plates and grown as reported below. After 24 h, 10 nM of each siRNA and universal scrambled negative controls were co-transfected with 2 μg of BRCA2-CT minigenes using Lipofectamine 2000 (Invitrogen), according to the manufacturer’s instructions.

In particular, gene-specific siRNA oligomers (20 μM) were diluted in a 250 μL Opti-MEM I reduced serum medium (Opti-MEM, Invitrogen) and mixed with 5 μL of each transfection reagent pre-diluted in 250 μL Opti-MEM. After 20 min of incubation at room temperature, the complexes were added to the cells in a final volume of 500 μL medium.

The RNAi results were evaluated by quantitative real-time PCR (qPCR) and western blot analysis 24 and 48 h after the transfection cells were harvested for RNA extraction, while after 48 h the proteins were collected for a western blot analysis (Appendix A).

#### 4.9.1. RT-PCR and qPCR

The total RNA was isolated with the Trizol reagent (Invitrogen), and then 2 μg of RNA were used for the reverse transcription (RT) reactions using the Super Script Vilo Master Mix (ThermoFisher Scientific, Waltham, MA, USA), following the manufacturer’s protocol. Subsequently, qPCR was used to quantify the mRNA expression levels of the target genes (BRCA2 and Actin) by using 3 μg of cDNA with SYBR Select Master Mix (ThermoFisher Scientific, Waltham, MA, USA). The PCRs were run with a 10 μL reaction volume containing 5 μL SYBR Select Master Mix, 10 μM each of primers. The PCR parameters were as follows: 95 °C for 10 min, 40 cycles through 95 °C for 15 s, 60 °C for 1 min. A melting curve analysis (from 65 to 95 °C, followed by cooling to 50 °C) was also performed to exclude non-specific PCR products. PCR analyses were conducted in triplicate for each sample. The reported values were calculated using the delta-delta Ct method and normalized against endogenous Actin. The primers were listed in Appendix A.

#### 4.9.2. Western Blot Analysis

Whole-cell protein extracts used for western blot analysis were collected as reported above. The proteins were separated on 6% polyacrylamide gel: the latter run was at 120 V for about 2 h, and the electrophoretic run was stopped when the 72 kDa blue marker left the gel. Then, it was transferred onto a polyvinylidene difluoride Immobilon P membrane (Bio-Rad) using the Trans-Blot Turbo Transfer System (Bio-Rad), with the Turbo High precast protocol. The filters were blocked with 5% milk-TBST (50 mMTris pH 7.5, 150 mM NaCl, 0.1% Triton X-100) for 1 h at room temperature. The BRCA2 endogenous proteins were immunodetected using the Anti-BRCA2 (Ab-1) Mouse mAb (2B) (OP-95 Calbiochem, Merck KGaA, Darmstadt, Germany; 1:300). Vinculin (SC-7649, Santa Cruz Biotechnology, CA, USA; 1:1000) was used as a loading control. Anti-mouse and anti-goat IgG conjugated to horseradish peroxidase was used for the secondary antibodies (GE Healthcare, Chicago, Ill, USA). The bands were detected with ECL chemiluminescence methods (PerkinElmer, Waltham, MA, USA) and by exposure to the ChemiDoc MP System (Bio-Rad). For all Western Blot figures, the whole blot results are shown in Appendix A). 

#### 4.9.3. In Vitro ssDNA Binding Assay on Silenced and Scrambled Negative Controls Samples

Forty-eight hours after the co-transfections, the cells were irradiated with 10 Gy and harvested 4 h later for nuclear proteins extractions, as described above. Then, 5 μg of the nuclear extracts of the silenced and non-silenced cells were added to 3 pmol of linear ϕX174 5′-Cy5-end-labeled ssDNA oligonucleotides, as previously described. The protein-DNA complexes were resolved, were run and were detected in the conditions described above.

### 4.10. RAD51 Immunofluorescence

U2OS cells were grown on coverslips and irradiated with 10 Gy 24 h after transfection. Four hours later, the cells were permeabilized by incubating them twice for six minutes on ice in CSK buffer (10 mM PIPES pH 7.0, 100 mM NaCl, 300 mM sucrose, 3 mM MgCl2, Triton X-100 0.7% and 0.3 mg/mL RNase A) [36]. From each transfection point, a non-permeabilized control coverslip was directly fixed with 4% paraformaldehyde. The coverslips were incubated with blocking buffer (10% FBS, 1% BSA, 0.2% Triton X-100 in phosphate buffer saline) for 15 min at room temperature, then incubated with the primary antibodies ANTI-FLAG^®^ M2, Clone M2 (1:1,000, F1804, Sigma-Aldrich) and anti-RAD51 H-92 (1:200; Santa Cruz Biotechnology Inc., TX, USA) overnight at 4 °C. After three washes with PBS, the coverslips were incubated with the secondary antibodies Alexa488-conjugated anti-mouse or the Alexa546-conjugated anti-rabbit (Molecular Probes, Eugene, OR, USA). The nuclei were stained with 2 μg/mL DAPI. Images were captured on a Zeiss confocal microscope, at 40× and 63×. The Rad 51 foci were analyzed using ImageJ software (National Institutes of Health (NIH, Bethesda, MA, USA) and Laboratory for Optical and Computational Instrumentation (LOCI, University of Wisconsin, Madison, WI, USA). At least 100 nuclei were counted; cells forming > 5 foci were scored as positive.

### 4.11. DNA Damage Recovery Assay

The NIH-3T3 cells were plated on coverslips and co-transfected with the BRCA2-CT cDNA constructs and pmaxGFP^®^ Vector (Lonza) (10:1 rate). Twenty-four hours after the transfection, the cells were irradiated with 5 Gy. The cells were fixed 30 min, 12 h, 24 h and 36 h after irradiation with 4% PFA for the immunofluorescence assay. A non-irradiated control was fixed for each experimental point. The coverslips were blocked and permeabilized using 10% FBS, 1% BSA, and 0.2% Triton X-100 in PBS 30 min at room temperature and incubated with anti-phospho-Histone H2A.X (Ser139) (1:300, Millipore, MA, USA) overnight at 4 °C. After washing, the coverslips were incubated with the Alexa Fluor 594-conjugated secondary antibody (Molecular Probes) for one hour at room temperature. DAPI (Molecular Probes) was used to stain the nuclei. Fluorescence was observed with the Leica DMS 4000B microscope (Leica, Wetzlar, Germany) with a 20× objective. DNA damage recovery was quantified as the number of γ-H2AX-positive cells on the total number of GFP-positive cells. Ten fields for each sample were analyzed with the ImageJ software, and the number of γ-H2AX-positive cells was calculated from a minimum of 500 cells per dose/time point.

### 4.12. Statistical Analysis

All experiments were repeated at least three times. The error bars are represented in terms of SD. Samples with a normal distribution were analyzed by the Student’s *t* test. The differences between groups were considered significant at *p* < 0.05. The denoted P-values are reported as follows: ns (*p* > 0.05), * (*p* ≤ 0.05), ** (*p* ≤ 0.01), *** (*p* ≤ 0.001), **** (*p* ≤ 0.0001).

## 5. Conclusions

To assess the functional significance of the p.Pro2767Ser VUS, we developed an experimental strategy by which truncated BRCA2 proteins were cloned in minigenes that encode the CT region, including the DBD and NLS domains. Thus, we were able to analyze the behavior of a mutated sequence that altered the BRCA2 DBD, and to demonstrate that it exerts a competitive effect on the normal physiological BRCA2 processes.

In conclusion, our study provides functional evidence of the potential pathogenicity of the p.Pro2767Ser variant and of its potential role in predisposing to HBOC. Importantly, as our patient’s phenotype is most probably a consequence of the BRCA2 p.Pro2767Ser variant, the latter should be included in the genetic testing of HBOC patients and their relatives, to assess their cancer risk. This study underlines the importance of studying the effects of sequence variants via functional evaluations to determine their potential role in disease onset, in patients’ risk stratification and in therapeutic decision-making.

## Figures and Tables

**Figure 1 cancers-11-01454-f001:**
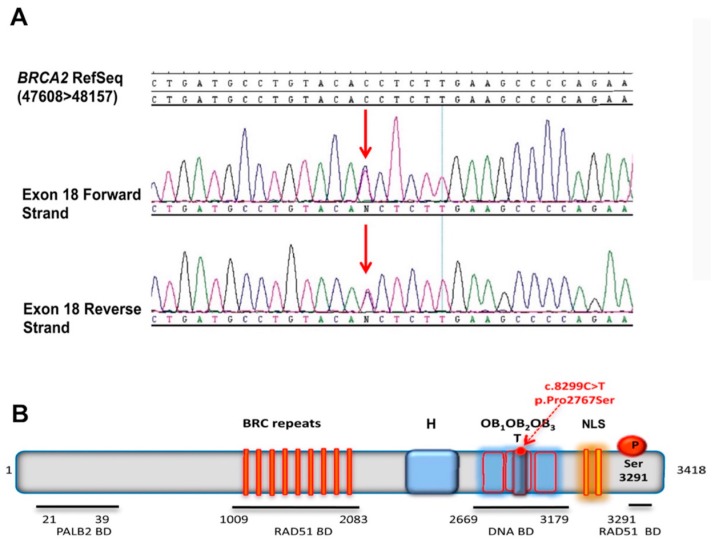
Sanger sequencing of the exon 18 of BRCA2 and BRCA2 full-length protein representation. (**A**) Sanger sequencing of the exon 18 of the BRCA2 gene performed on the patient’s DNA showing the c.8299C > T p.Pro2767Ser variant in the heterozygous state on forward and reverse strands. (**B**) Schematic representation of the full-length BRCA2 protein (3418 aa) with its functional domains. The N-terminal domain (aa 10–39) includes the PALB2 binding domain (PALB2 BD), the eight BRC repeats (aa 1009–2082) that are necessary for RAD51 monomer binding (RAD51 BD), the DNA binding domain (DBD) in the C-terminal region constituted by a helical (H) domain (aa 2402–2668), three OB folds that bind to single-strand DNA (aa 2669–3102), and a tower domain (T) that binds to double-strand DNA; the C-terminal end contains an additional RAD51 binding site (aa 3270–3305) activated by Serine 3291 phosphorylation.

**Figure 2 cancers-11-01454-f002:**
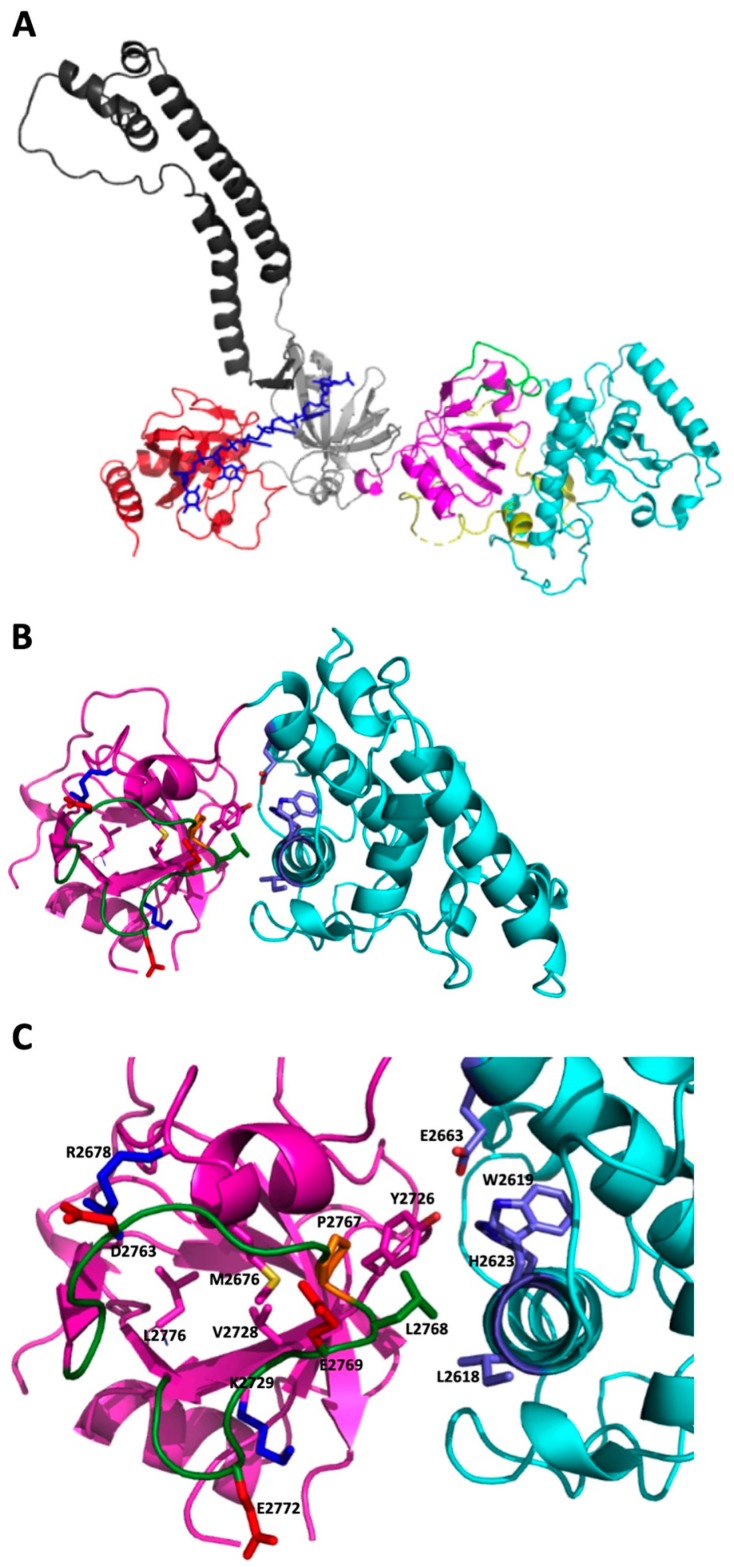
The 3D modeling analysis of the BRCA2 DBD. (**A**) A schematic representation of the structural model for the human BRCA2 DBD domain. N-terminal Helical domain (cyan); OB1 domain (magenta; L45 loop in green; the coiled-coil region L12 between β1 and β2 has been omitted from the model); OB2 domain (grey with the tower-like structure in dark grey); OB3 (red); ssDNA 6 nucleotides (blue); DSS1 protein fragment (yellow). The positions of ssDNA and DSS1 are derived from the 1MJE and 1MIU PDB structures. (**B**) A schematic representation of the H (cyan) and OB1 (magenta) domains. The loop L45 is shown in green. In stick representation, (**C**) the side chains of the following residues are represented: in the L45 loop, p.Pro2767 is depicted in orange, p.L2768 in green and the three acidic residues in red (p.D2763, p.E2769, and p.E2772). In the OB1 domain, p.Y2726, p.V2728, p.M2676 and p.L2776 are in magenta, whereas the basic residues p.R2678 and p.K2729 are shown in blue. In the H domain, p.E2663, p.W2619, p.H2623, and p.L2618 are shown in light blue. The secondary structure nomenclature refers to Yang et al. [24].

**Figure 3 cancers-11-01454-f003:**
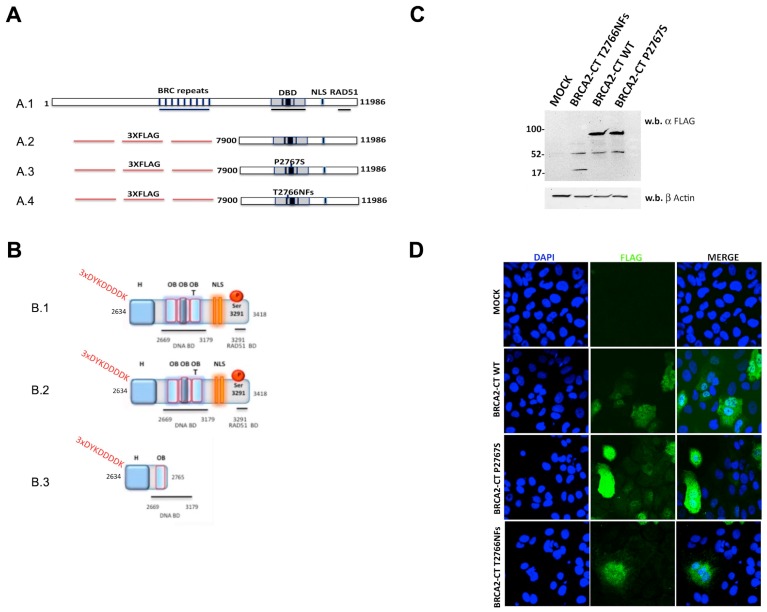
BRCA2 minigenes and proteins expression. (**A**) Schematic representations of the BRCA2 minigenes. (**A.1**) The wild-type full-length 11986bp-BRCA2 cDNA. (**A.2**) The wild-type BRCA2 minigene encoding the C-terminal (CT) region of the BRCA2 protein. (**A.3**) The mutagenized BRCA2 minigene carrying the c.8299C > T sequence variant encoding the p.Pro2767Ser variant DNA binding domain (DBD). (**A.4**) The mutagenized BRCA2 minigene carrying the c.8297delC variant and encoding a truncated CT region lacking the DBD. Sequences encoding 3XFLAG were inserted at the N-terminal region. (**B**) A schematic representation of the cloned BRCA2-CT proteins. (**B.1**) The cloned BRCA2-CT wt protein contains only the C-terminus domain of the BRCA2 protein. It ranges from aa 2634 to aa 3418, and contains part of the helical (H) domain (aa 2402–2668), three OB folds and the tower domain (T), the nuclear localization sequences (NLS), and the additional RAD51 binding domain (RAD51 BD). 3XFLAG is expressed at the N-terminal domain of this protein. (**B.2**) BRCA2-CT P2767S contains the CT domain of the BRCA2 protein, as described above. The p.Pro2767Ser variant falls in the T domain. (**B.3**) BRCA2-CT T2766NFs is a truncated protein derived from a premature stop codon insertion. It ranges from aa 2634 to aa 2765 and contains part of the H domain (aa 2402–2668) and only one of the three OB folds. It does not contain the NLS or any RAD51 binding domains. The c.8297delC variant falls in the OB1 fold. (**C**) A western blot analysis of the expression of BRCA2-CT wt, BRCA2-CT P2767S and BRCA2-CT T2766NFs proteins in NIH-3T3 cells. A mutant BRCA2 protein of 95 kDa was detected in cells that contained BRCA2-CT wt and BRCA2-CT P2767S proteins; a 15 kDa deleted BRCA2-CT T2766NFs protein derived from the Thr2766AsnFs mutation. (**D**) The nuclear localization of the BRCA2-CT wt, BRCA2-CT P2767S and the BRCA2-CT T2766NFs proteins in U2OS cells stained with the anti-FLAG antibody and analyzed by confocal microscopy. Scale bars 0.44 μm.

**Figure 4 cancers-11-01454-f004:**
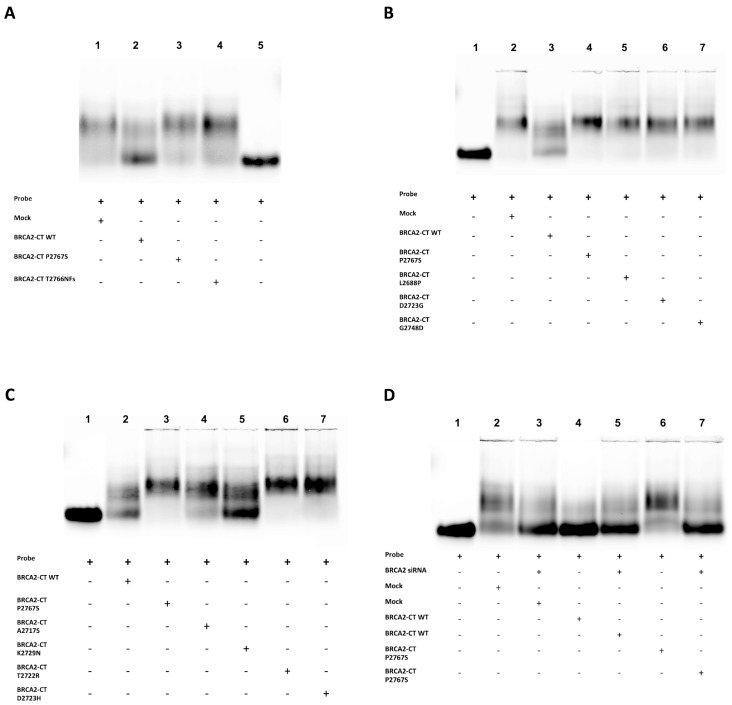
The p.Pro2767Ser likely pathogenic variant hampers the binding of BRCA2 to DNA. (**A**) The ssDNA binding assay of nuclear extracts from 10 Gγ-IR mock-transfected cells and cells expressing the BRCA2-CT wt, and the BRCA2-CT P2767S and BRCA2-CT T2766NFs proteins. In the mock sample, the endogenous BRCA2-RAD51 complex was bound to ssDNA. The electrophoretic migration was delayed in the BRCA2-CT P2767S, and the BRCA2-CT T2766NFs samples, but not in the BRCA2-CT wt sample. (**B**) The ssDNA-binding assay of BRCA2-CT WT and BRCA2-CT P2767S compared to the pathogenic BRCA2-CT controls. BRCA2-CT P2767S showed a similar migration delay compared to all the pathogenic BRCA2-CT controls. (**C**) The 2 benign controls shifted similarly to BRCA2-CT WT, while the remaining pathogenic controls behaved similar to BRCA2-CT P2767S, in which endogenous BRCA2 can act properly and binds ssDNA after irradiation. Due to the dimension of the electrophoretic chamber, up to seven samples per run were loaded. (**D**) The effect of the endogenous BRCA2 silencing assessed by the in vitro ssDNA-binding assay on silenced and non-silenced samples 48 h after transfections. In the BRCA2-silenced-cells, there were no differences of migration pattern among the samples.

**Figure 5 cancers-11-01454-f005:**
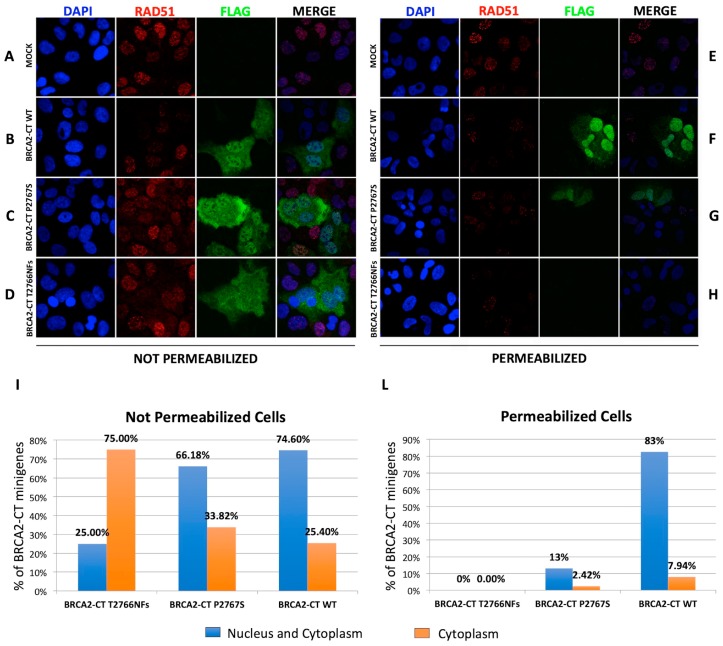
The analysis of the binding of nuclear BRCA2-CT proteins to chromatin by immunofluorescence in U2OS cells. (**A**) Not permeabilized U2OS mock cells and U2OS cells transfected with (**B**) BRCA2-CT wt, (**C**) BRCA2-CT P2767S and (**D**) the BRCA2-CT T2766NFs minigenes. The nuclear localization of ectopically expressed FLAG-tagged proteins is shown in green. Scale bars: 0.22 μm. (**F**) Permeabilized mock cells did not generate any signal. Conversely, in permeabilized U2OS cells transfected with (**F**) BRCA2-CT wt, with (**G**) BRCA2-CT P2767S and with (**H**) the BRCA2-CT T2766NFs minigenes, only the BRCA2-CT wt protein was detected, whereas the BRCA2-CT P2767S and BRCA2-CT T2766NFs proteins were predominantly washed out during the permeabilization procedure. (**I**) Percentage of transfected cells showing predominantly nuclear, nuclear and cytoplasmic and cytoplasmic-only BRCA2-CT staining before the permeabilization procedure. In Figure 5I,L the Y-axis represents the percentage of BRCA2-CT minigenes that localize predominantly both in the nucleus and in the cytosol or only in the cytosol. All constructs showed a principally nuclear and cytoplasmic localization. In particular, without permeabilization, the BRCA2-CT WT and the BRCA2-CT P2767S predominantly localized in the nucleus (74.60% and 66.18% respectively), while the BRCA2-CT T2766NFs was principally in the cytoplasm (75% with respect to 25% of nuclear staining). (**L**) The quantification of BRCA2-CT proteins localization, after the permeabilization procedure, calculated on the total FLAG-expressed cells for each BRCA2-CT sample. Prevalently, 83% of the BRCA2-CT WT protein was nuclear-localized, while only 13% of the BRCA2-CT P2767S was found in the nucleus, and no BRCA2-CT T2766NFs was detected in either the nucleus or the cytoplasm.

**Figure 6 cancers-11-01454-f006:**
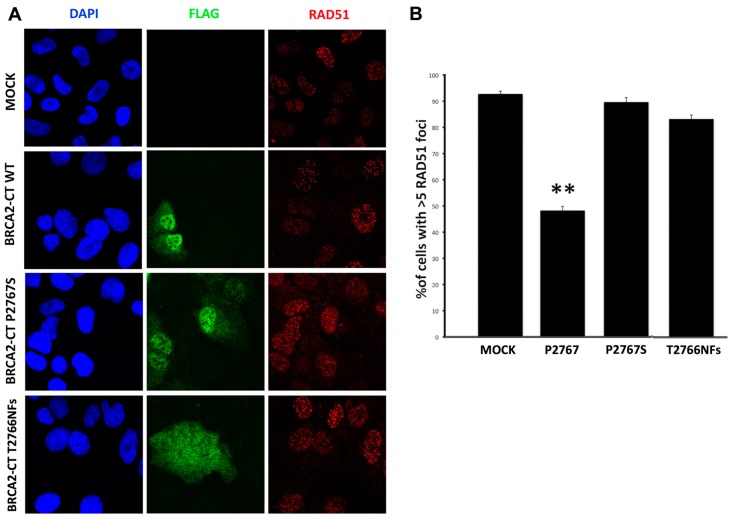
The analysis of the RAD51 foci formation. (**A**) The RAD51 foci formation assay upon DNA damage, 4 h post-irradiation (IR). The presence of RAD51 foci (red) in the cell nuclei shows the activation of the DNA damage repair pathway after IR. BRCA2-CT P2767S-expressing and BRCA2-CT T2766NFs-expressing cells form more RAD51 foci than BRCA2-CT wt cells do. Scale bars: 0.22 μm. (**B**) The RAD51 foci count in the transfected cells, 4 h post-IR. The percentage of RAD51 foci on the FLAG-positive cells was significantly higher in the BRCA2-CT P2767S cells than for BRCA2-CT wt (*p* ≤ 0.001) and was comparable to the mock cells (*p* ≤ 0.01).

**Figure 7 cancers-11-01454-f007:**
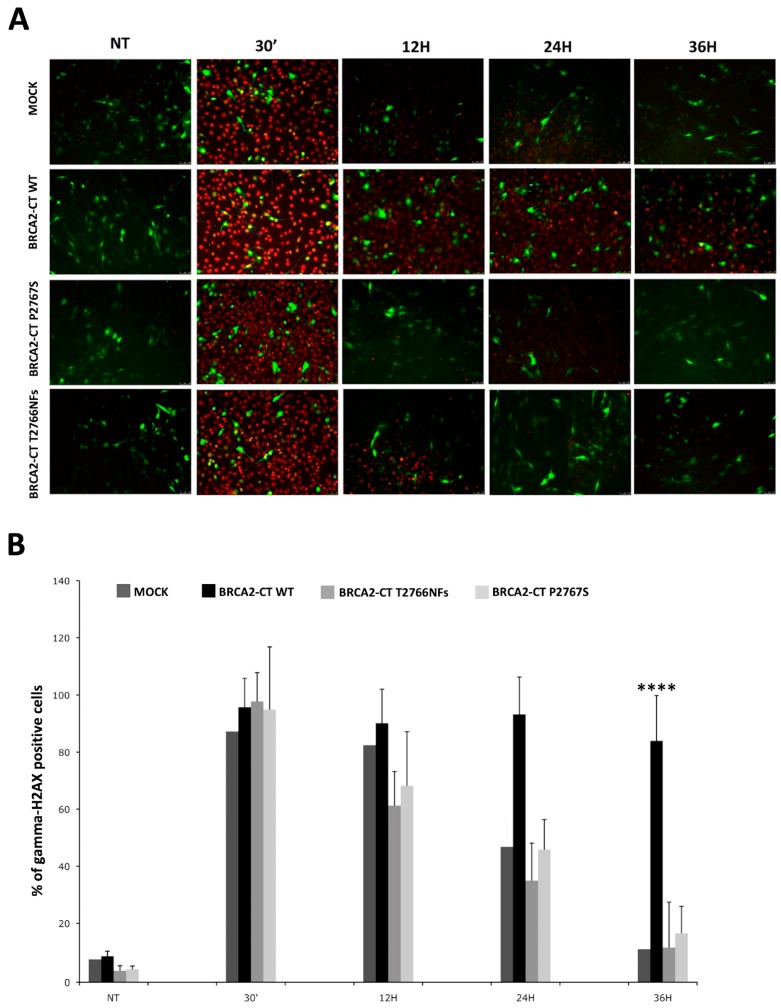
The DNA damage recovery assay. (**A**) Phospho-H2AX-recovery kinetics representative figures. GFP (green) and phospho-H2AX (red) emission in transfected cells at time 0 (untreated, NT), at 30 min, 12 h, 24 h and 36 h after 5 Gy/sec irradiation (IR). At least 500 GFP/γH2AX-positive cells were counted for each sample. Scale bars 50 μm. (**B**) The percentage of the phospho-H2AX count. All NT cells at time 0 showed basal levels of histone H2AX phosphorylation: Mock 7.6%, BRCA2-CT wt 8.6%, BRCA2-CT P2767S 4.3% and BRCA2-CT T2766NFs 3.8%. Thirty minutes after IR, the percentage of H2AX phosphorylation was: mock 87.2%, BRCA2-CT wt 95.7%, BRCA2-CT P2767S 94.8% and BRCA2-CT T2766NFs 97.7%. Twelve hours after IR, the values were: Mock 82.4%, BRCA2-CT wt 90.1%, BRCA2-CT P2767S 68.1% and BRCA2-CT T2766NFs 61.2%. At 24 h, BRCA2-CT P2767S showed 45.7% of H2AX phosphorylation, BRCA2-CT T2766NFs 34.9% and mock 46.7%. The DNA of the cells transfected with BRCA2-CT wt showed 93.2% of phospho-H2AX. The percentage of γ-H2AX positive cells at 36 h after IR was 16.5% in BRCA2-CT P2767S, 11.6% in BRCA2-CT T2766NFs and 11% in the mock cells but remained very high in the BRCA2-CT wt cells, namely 83.9% of H2AX phosphorylation (**** = 2.4 × 10^−11^; *p* ≤ 0.0001). Differences between BRCA2-CT P2767S, BRCA2-CT T2766NFs and the mock-transfected samples were not statistically significant.

**Figure 8 cancers-11-01454-f008:**
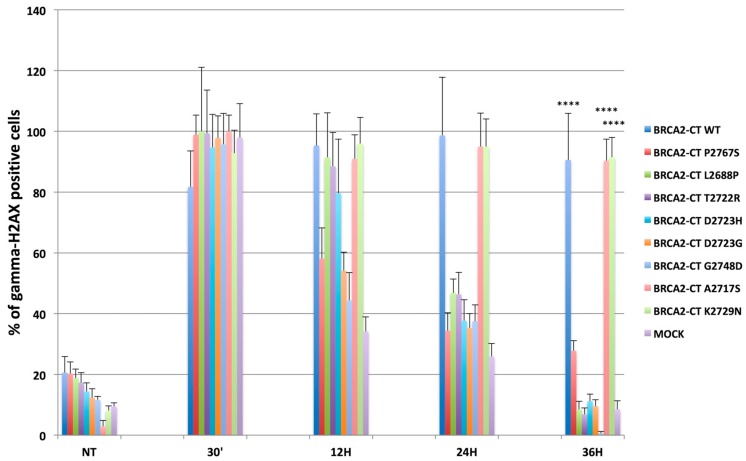
The DNA damage recovery assay performed on the entire variants validation set. The percentage of gamma-H2AX positive cells, before 5Gγ irradiation, showed a basal level of H2AX phosphorylation for all samples. Thirty minutes after IR, each sample showed high levels of γ-H2AX that started to decrease, at different rates, until 36 h. With respect to BRCA2-CT P2767S, the DNA of the cells transfected with the BRCA2-CT WT minigene (90.6%, **** = 2.34 × 10^−17^, *p* ≤ 0.0001) and the BRCA2-CT benign controls (BRCA2-CT A2717S, **** = 9.8 × 10^−33^, *p* ≤ 0.0001, and BRCA2-CT K2729N, **** = 2.4 × 10^−32^, *p* ≤ 0.0001) after 36 h post-IR, was not repaired, unlike in BRCA2-CT P2767S (27.76%, *p* ≤ 0.0001), BRCA2-CT L2688P (17.14%, **** = 1.4 × 10^−16^, *p* ≤ 0.0001), BRCA2-CT T2722R (11.43%, **** = 2.74 × 10^−16^, *p* ≤ 0.0001), BRCA2-CT D2723H (10,81%, **** = 6.25 × 10^−16^, *p* ≤ 0.0001), BRCA2-CT D2723G (12%, **** = 1.4 × 10^−17^, *p* ≤ 0.0001), BRCA2-CT G2748D (totally repaired, 0%, **** = 8.9 × 10^−16^, *p* ≤ 0.0001), and the mock-transfected cells (8.47%, **** = 16.9 × 10^−17^, *p* ≤ 0.0001). Instead, the decrease of γ-H2AX-positive signals in the cells transfected with BRCA2-CT P2767S and all the BRCA2-CT pathogenic controls minigenes (BRCA2-CT L2688P, BRCA2-CT T2722R, BRCA2-CT D2723H, BRCA2-CT D2723G, and BRCA2-CT G2748D) were similar to the mock-transfected cells, and differences regarding these samples were not statistically significant.

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
