# Peer review of "A Functional Analysis of the Unclassified Pro2767Ser BRCA2 Variant Reveals Its Potential Pathogenicity that Acts by Hampering DNA Binding and Homology-Mediated DNA Repair"

_cancers, 2019, doi:10.3390/cancers11101454_

Round 1

Reviewer 1 Report

In this manuscript, Esposito et al. report the functional characterization of a BRCA2 variant of unknown clinical significance, c.8299C>T, which results in a proline to serine substitution at codon 2767.  This variant was identified in a patient who was diagnosed of breast cancer at the age of 26.  The authors performed in silico analysis and 3D structure prediction and concluded that that the structure of the DNA binding domain is affected by the amino acid substitution.  The authors have used a minigene construct expressing a truncated BRCA2 protein to examine the effect of the variant on its ability to bind to ssDNA and displace endogenous BRCA2.  They have compared the impact of P2767S with WT and a number of known neutral and pathogenic variants.  The impact on DNA binding was validated using a cell permeability assay, which allows displacement of protein that is not bound to the chromatin.  While the WT protein is detected by IF in the nucleus, S2767 and T2766NFs were barely detectable consistent with the results of the in vitro DNA binding assay. In addition, the effect on IR-induced RAD51 foci formation and γ-H2AX staining was examined. Overall the findings are interesting, and the study provides evidence to support the pathogenic nature of P2767S variant.  However, there are some concerns that should be addressed.

Major concerns

The in vitro DNA binding results are not very convincing. It is a valid and novel approach to determine the functional significance of BRCA2 unclassified variants. However, it is unclear what the ssDNA probes are binding to in the nuclear extract. The effect of irradiation and transfection on binding of the ssDNA to the probe is also unclear. A dose dependent binding assay should be performed using increasing amount of the nuclear extract to show that the DNA-protein complex formation increases when more protein is present. No data has been provided to support that the minigene products are displacing endogenous BRCA2.  BRCA2 knockdown or use of BRCA2 deficient cells can help address some of these concerns. Antibody-mediated super-shift would also show specific interaction between the probe and BRCA2. Quantification of RAD51 foci shown in Figure 6B does not exactly reflect the impact of BRCA2-CT WT, P2767S and P2766NFs shown in Figure 6A. Even though the number of foci in GFP positive cells expressing WT is fewer than those in cells expressing the BRCA2 mutants, there is comparable reduction in the intensity of the foci in all cases especially in cells expressing P2766NFs.

Minor comments:

Figure 3D hardly shows BRCA2-CT WT in the nucleus. There is no difference in the staining between WT and T2766NFs.

Figure 5I and L, what is represented on the Y axis? If it is percentage of BRCA2 protein in the nucleus vs. cytoplasm, how were these values calculated for “permeabilized” samples?

Author Response

Q: The in vitro DNA binding results are not very convincing. It is a valid and novel approach to determine the functional significance of BRCA2 unclassified variants. However, it is unclear what the ssDNA probes are binding to in the nuclear extract.

The effect of irradiation and transfection on binding of the ssDNA to the probe is also unclear. A dose dependent binding assay should be performed using increasing amount of the nuclear extract to show that the DNA-protein complex formation increases when more protein is present.

A: We thank the Reviewer for the useful suggestions that allow us to improve some aspect of our paper.

In this study we demonstrated that the BRCA2-CT WT and other benign controls, behave differently from pathogenic controls and the BRCA2-CT P2767S (the mutated newly found protein region) only when cells are irradiated thereby demonstrating that a more precise stimulus was required to activate the cells, as occurs in the case of the DNA double strand break. Indeed, DNA-repair activated proteins were able to bind the ssDNA probe differently when activated by DNA damage.

To address the Reviewer’s point, a dose dependent binding assay was performed for a BRCA2-CT pathogenic control (BRCA2-CT L2688P) and for a BRCA2-CT benign control (BRCA2-CT K2729N). Therefore, we performed the ssDNA binding assay using increasing amount of the nuclear extract: for each sample we loaded the binding reaction with only free probe (as 0 micrograms of nuclear extract), 1,5 μg, 2,5 μg, 3,5 μg, 5 μg and 6,5 μg of nuclear extract of irradiated cells.

As described in the Result section (page 6, lines 345-349) and showed in Supplementary Figure S5 A and B (lane 1 up 6), and commented in Figure S5 legends, the DNA-protein complex formation increases when more protein is present. In particular, differences between BRCA2-CT L2688P and BRCA2-CT K2729N are clearly shown when protein is increased and particularly in lane 5 and 6, from 5 up to 6,5 μg of nuclear extract, giving answer to the question posed.

Q: No data has been provided to support that the minigene products are displacing endogenous BRCA2. BRCA2 knockdown or use of BRCA2 deficient cells can help address some of these concerns. Antibody-mediated super-shift would also show specific interaction between the probe and BRCA2.

A: We take the Reviewer’s point; in our paper we speculate by stating that our BRCA2-CT minigenes could compete with endogenous BRCA2 because of their BRCA2 DBD region and their capability to bind ssDNA when overexpressed in cells in which DNA DSBR is activated. However, to clarify this point, now we have prepared a BRCA2-siRNA system to deplete the endogenous BRCA2 and to analyze the behavior of our BRCA2-CT minigenes in its absence.

Therefore, we performed a BRCA2 silencing by using the Brca2 - 3 unique 27mer siRNA duplexes in NIH-3T3 cells. Then, 48hr after siRNA transfection, cells were irradiated and harvested to isolate nuclear proteins; consequently, we assessed the effects on our BRCA2-CT minigenes by performing the ssDNA binding assay when the endogenous BRCA2 was silenced, in the same conditions of those described in the manuscript (lines 851-864, page 19). Nuclear extracts from 10 Gγ irradiated and BRCA2-silenced NIH-3T3 cells were incubated with 100 nt-long Cy5-labelled Ï•X174 ssDNA, and the proteins-DNA complexes were analyzed by electrophoresis (new Figure 4D). Interestingly, in cells where endogenous BRCA2 is silenced, there are no differences in behavior among samples because the main assumption of these differences is missing: there is no competition with endogenous BRCA2 protein and there are no differences among samples (new Figure 4D, lanes 3, 5 and 7). Indeed, in not-silenced cells, endogenous BRCA2 is properly expressed and BRCA2-CT minigenes showed a different shift pattern (Figure 4D, lane 2, 4 and 6) according to their capability to compete with endogenous BRCA2 and to bind to the Ï•X174 ssDNA probe.

All silencing experimental details are described in Materials and Methods (lines 865-908, pages 19-20) and in the Supplementary Section (Figure S6A and B and legends); experimental results are described in the Results section (lines 402-414, page 8) and in the new Figure 4D and Figure S6A and B.

Q: Quantification of RAD51 foci shown in Figure 6B does not exactly reflect the impact of BRCA2- CT WT, P2767S and P2766NFs shown in Figure 6A. Even though the number of foci in GFP positive cells expressing WT is fewer than those in cells expressing the BRCA2 mutants, there is comparable reduction in the intensity of the foci in all cases especially in cells expressing P2766NFs.

A: The figure 6A is representative of several acquisitions on which RAD51 foci were counted. However, the reduction of the RAD51 foci intensity of cells expressing the BRCA2-T2766NFs is lower than BRCA2-CT WT transfected cells but greater either in BRCA2-CT P2767S or in mock-transfected cells. The percentage of RAD51 foci on FLAG-positive cells was significantly higher in BRCA2-CT P2767S cells versus BRCA2-CT wt (P ≤0.001) and comparable to mock cells. According to Reviewer’s observation, now we clarify this point in the text at page 11 at lines 489-490; 495-500 and in the Figure 6 legend (line 511).

Minor comments:

Q: Figure 3D hardly shows BRCA2-CT WT in the nucleus. There is no difference in the staining between WT and T2766NFs.

A: We provide now an high-resolution 3D figure to improve visualization. We also slightly reviewed the text at page 6 at lines 329-330.

Q: Figure 5I and L, what is represented on the Y axis? If it is percentage of BRCA2 protein in the nucleus vs. cytoplasm, how were these values calculated for “permeabilized” samples?

A: We thank the Reviewer’s question that allows us to improve our Figure. On the Y-axes of Figure 5I and L are represented the percentage of BRCA2-CT minigenes. Regarding the second point raised by the reviewer, because the permeabilized cells derive from slides of the same transfection plate, for each BRCA2-CT sample, is used as “absolute value” the total number of FLAG-expressed cells before the permeabilization procedures. The manuscript has been changed accordingly (at page 10, lines 448-451 and 456-458) in the Figure 5 legend (at page 11, lines 474-476; 481-482).

Reviewer 2 Report

In this manuscript, Esposito et al. provide an analysis of the BRCA2 variant p.P2767S, currently classified as a variant of uncertain clinical significance. In the absence of definitive family data to determine pathogenicity, the authors present in silico predictions and a functional assay to determine its impact on function. The authors develop an indirect approach in which a fragment of the BRCA2 C-terminal is ectopically expressed and is expected to compete with endogenous BRCA2 functions (as measured by subcellular localization, RAD51 focus formation and gamma-H2AX levels). When a variant leads to a functional impact on the protein, a fragment containing this variant is then expected not to compete with endogenous BRCA2. The assay is an interesting addition to the functional analysis literature and the work was carefully conducted.

Although the assay is novel and includes appropriate controls, it is somewhat disappointing that the authors only evaluate one VUS. When the field is usually producing analyses of hundreds and sometimes thousands of variants, publication of an analysis for a single variant seems problematic to me unless there is something compelling, informative, or generalizable about this variant or the method.

The authors should provide a justification for the choice of bioinformatics predictors used among more than 40 currently available, and which parameters were used.

It is unclear to me the value of modelling analysis part of the manuscript. It is a rather lengthy part with some structural insights. I wonder if this part would best fit in the discussion as a part of a rationalization for the defective function of the variant.

The discussion and conclusion should be revised. For example, page 16 seems a re-hash of the results section. But most importantly the evidence to link the variant to the carrier phenotype is extremely weak.  The discussion should include a description of the limitations (such as not testing for other genes besides BRCA1/2), and a careful consideration of the potential limitation of the functional assay. In particular, while the authors can make the case that the variant affects BRCA2 function, and suggest that it could constitute a pathogenic variant, without additional data this variant should not be considered pathogenic or likely pathogenic for clinical decisions. This may seem self-evident but the paper should state very clearly to avoid misuse of this information in the clinical setting.

Minor issues:

3D FLAG figure for the P2767 seems shifted when compared to the merged Why aren’t all variants represented in the same gel in 4B&C? Please italicize human gene symbols, use HGVS nomenclature throughout (p.P2767S not P2767S), and change ‘mutation’ to ‘(pathogenic or likely pathogenic) variant’ to align with recent nomenclature recommendations (J Med Genet. 2019 Jun;56(6):347-357). Line 63 change step to steps Line 94 change evaluated to evaluate Line 90 should specify which NCBI database. Presumably ClinVar? Line 579 1 x 10E7 presumably not 1 x 107 Why Figure 7? and not only Figure 8 that contains all variants.

Author Response

Q: Although the assay is novel and includes appropriate controls, it is somewhat disappointing that the authors only evaluate one VUS. When the field is usually producing analyses of hundreds and sometimes thousands of variants, publication of an analysis for a single variant seems problematic to me unless there is something compelling, informative, or generalizable about this variant or the method.

A: The essential reason is that it took a long time to assess this method and we also thought that this method, when published, would stimulate other groups, which have other case, or family with similar situation that may try to use it to extend the validity of the method. We now put this limitation toward the final part of the paper.

Moreover, we evaluated only one VUS because of different reasons. The first is that this variant was found in a young woman that underwent BRCA1/2 NGS screening. In particular, due to the young age of onset (26 years) and to the echography appearance showing several highly vascularized hypoechoic area with numerous calcifications, she underwent surgery and was admitted to an under-forty surveillance program. Given the young age of the patient and given no other pathogenic mutations were found, we characterize its role in disease onset and to enable clinicians to provide better counseling, follow-up and cancer-risk assessment in relatives.

Second, we described a novel method with which to assess the role of variants that fall in the DBD of BRCA2 useful to overcome the cumbersome and complex assays in which the entire BRCA2 protein should be analyzed. In this context, the method we report herein is much faster than “traditional” functional assay and consequently results in faster clinical decisions for patient health. We now clarify this point in the Discussion section at page 15 lines 605-614.

Q: The authors should provide a justification for the choice of bioinformatics predictors used among more than 40 currently available, and which parameters were used.

A: Sometimes, the use of too many prediction tools can give conflicting results and this is the reason why we used the SIFT and PolyPhen-2 tools as first evaluation. Then, evolutionary conservation of the variant was analyzed using Align-GVGD and then we use the Mutation Taster tool. Some of these tools are included in the VarSome website, which allowed us to use also other 14 bioinformatic predictors: all agreed with the prediction of pathogenicity. All software were used with the default parameters. We now specify this point in the Material and Methods section at page 17, lines 774-775.

Q: It is unclear to me the value of modelling analysis part of the manuscript. It is a rather lengthy part with some structural insights. I wonder if this part would best fit in the discussion as a part of a rationalization for the defective function of the variant.

A: We performed modeling analysis to better characterize the effect of P2767S variant on BRCA2 DBD tridimensional structure, stability during DNA binding and conformation. We have moved now a large portion of the text to the Discussion section inserted at page 16, lines 644-668.

Q: The discussion and conclusion should be revised. For example, page 16 seems a re-hash of the results section.

A: Done, see page 16; some periods as indicated by the corrections have been eliminated or shortened.

Q: But most importantly the evidence to link the variant to the carrier phenotype is extremely weak.

The discussion should include a description of the limitations (such as not testing for other genes besides BRCA1/2), and a careful consideration of the potential limitation of the functional assay. In particular, while the authors can make the case that the variant affects BRCA2 function, and suggest that it could constitute a pathogenic variant, without additional data this variant should not be considered pathogenic or likely pathogenic for clinical decisions. This may seem self-evident but the paper should state very clearly to avoid misuse of this information in the clinical setting.

A: We take the Reviewer’s point and we clarify this part and the limitation by tuning down in the Discussion section at page 16-17, lines 685-746.

Minor issues:

Q: 3D FLAG figure for the P2767 seems shifted when compared to the merged. Why aren’t all variants represented in the same gel in 4B&C?

A: The reason is simply technical. Only the loading of up to seven samples per run was allowed in our electrophoretic chamber. We specify this point in the Figure 4 legend (page 9, lines 426-427).

Q: Please italicize human gene symbols, use HGVS nomenclature throughout (p.P2767S not P2767S), and change ‘mutation’ to ‘(pathogenic or likely pathogenic) variant’ to align with recent nomenclature recommendations (J Med Genet. 2019 Jun;56(6):347-357).

A: Done for both the observations.

Q: Line 63 change step to steps Line 94 change evaluated to evaluate Line 90 should specify which NCBI database. Presumably ClinVar?

A: We now clarify this point in the text, page 2 line 90.

Q: Line 579 1 x 10E7 presumably not 1 x 107 (Page 19 at line 653) Why Figure 7? and not only Figure 8 that contains all variants.

A: The experiment of Figure 7 is a different one from Figure 8, since it is a complementary quantification to the Figure 7. We now stress this point at page 14, lines 548-549.

Round 2

Reviewer 1 Report

The authors have satisfactorily addressed my concerns.